

# Pecan agroforestry systems improve soil quality by stimulating enzyme activity

Zhaocheng Wang[1,*], Mengyu Zhou[1,*], Hua Liu[1], Cheng Huang[1], Yuhua Ma[1], Hao xin Ge[2], Xiang Ge[2] and Songling Fu[1]

[1] School of Forestry and Landscape Architecture, Anhui Agricultural University, Hefei, Anhui, China
[2] Fuyang Xinfeng Seed Industry Co., Ltd., Fuyang, Anhui, China
[*] These authors contributed equally to this work.

## ABSTRACT

**Background**. Forest and plantation intercropping are efficient agroforestry systems that optimize land use and promote agroforestry around the world. However, diverse agroforestry systems on the same upper-plantation differently affect the physical and chemical properties of the soil.

**Methods**. The treatments for this study included a single cultivation (CK) pecan control and three agroforestry systems (pecan + *Paeonia suffruticosa* + *Hemerocallis citrina*, pecan + *Paeonia suffruticosa*, and pecan + *Paeonia lactiflora*). Soil samples were categorized according to the sampling depth (0–20 cm, 20–40 cm, 40–60 cm).

**Results**. The results demonstrated that the bulk density (BD) of soil under the pecan agroforestry system (PPH and PPL) was reduced by 16.13% and 7.10%, respectively, and the soil moisture content (MC) and total soil porosity (TPO) increased. Improvements in the physical properties of the soil under the PPS agroforestry system were not obvious when compared with the pecan monoculture. The soil total phosphorus (TP), total nitrogen (TN), available potassium (AK), and total carbon (TC) increased significantly, while the soil urease (S-UE), alkaline phosphatase (S-AKP), and 1,4-β-N-acetylglucosamines (S-NAG) enzyme activity also increased significantly, following agroforestry. Overall, the pecan agroforestry system significantly improved the physical properties of the pecan plantation soil, enriched the soil nutrients, and increased the activity of soil enzymes related to TC, TN, and TP cycles.

## INTRODUCTION

Soil is foundational for terrestrial ecosystems and its cumulative functions and microbial characteristics have the potential to alter environments on a global scale (*Rillig et al., 2019*; *Wilson & Lovell, 2016*). The physicochemical properties (*Chen et al., 2019*) and enzyme activity (*Wang et al., 2017*) of soils are typically considered to be important indicators of soil quality (*Paz-Ferreiro & Fu, 2013*). Soil enzymes participate in the decomposition of soil organic matter and play a key catalytic role. Organic matter is decomposed into nutrients for plants and improve the quality of the soil (*Ren et al., 2016*). The properties of soil, including the availability of nutrients and enzyme activity, typically vary with soil depth and can lead to inconsistent topsoil quality and subsoil factors (*Yang et al., 2021*).

Corresponding author
Songling Fu, fusongling@ahau.edu.cn

Agroforestry is a sustainable land management system used to maintain soil fertility and productivity (*Dollinger & Jose, 2018*; *Isbell et al., 2017*). The combination of tree and crop systems can optimize planting areas more effectively than individual tree systems (*Torralba et al., 2016*). It can promote the creation of jobs and generate income while protecting biodiversity and ecosystem services (*Muchane et al., 2020*; *Santos, Crouzeilles & Sansevero, 2019*). Many studies have shown that, in contrast to monocultures, agroforestry intercropping may promote a variety of agroecosystem services by increasing yields and improving soil quality and soil carbon sequestration (*Kimura et al., 2018*; *Ma et al., 2017*; *Paul, Sekhon & Sharma, 2018*; *Žalac et al., 2021*; *Zhang et al., 2021*). Agroforestry may also increase the content of soil organic carbon and total nitrogen (*Lian et al., 2019*; *Lu et al., 2015*). Improvements have also been shown in the physical properties (*Chen et al., 2019*), levels of available soil nutrients (N and K), and enzyme content (urease and acid phosphatase) of the soil (*Tang et al., 2020*).

Pecan (*Carya illinoinensis*) is a valuable nut tree popular in China (*Sagaram, Lombardini & Grauke, 2007*). Anhui Province was the first area to introduce pecans, and it has since become one of the main cultivation areas in China (*Zhang, Peng & Li, 2015*). In recent years, cultivation techniques (*Luo et al., 2016*), nut quality (*Atanasov et al., 2018*)), the chemical constituents of plant fruits (*Fernandes et al., 2017*), and the components and utilization of fruit shells (*Martinez-Casillas et al., 2019*)) have been studied to optimize the economy of the pecan. Pecan trees require large cultivation areas and extended growing periods (*Zhang, Peng & Li, 2015*). Woody crops, including peony, which is used for oil, herbaceous peony with medicinal value, and day lily are widely planted across northern subtropical China for their ecological and economic profitability (*Yang, Zhang & Li, 2018*). Pecan plantations are typically intercropped with other cash crops in agroforestry ecosystems, including Salvia miltiorrhiza and Arachis hypogaea. The transition from pecan monocultures to agroforestry systems is based on the ability of pecans to improve soil quality, soil fertility, and to improve the sustainability of farmlands (*Gao et al., 2019*; *Sagastuy & Krause, 2019*).

There have been few studies on the interactions between soil nutrients and soil enzyme activity under different agroforestry patterns of *Carya cathayensis* in northern subtropical China. We believe that the agroforestry management of pecans impacts soil quality. We studied three agroforestry systems (pecan + *Paeonia suffruticosa* + *Hemerocallis citrina* (PPH), pecan + *Paeonia suffruticosa* (PPS), and pecan + *Paeonia lactiflora* (PPL)), and a pecan monoculture (CK) to understand the effects of pecan compounds on soil quality (particularly its physical and chemical properties and enzyme activity).

We hypothesized that: the soil structure of pecan monocultures would be poor with low nutrient content; different agroforestry systems may improve the physical properties of the soil, optimize its structure, and enrich its nutrients to a certain extent; and the enzyme activity in the soils of different pecan agroforestry intercropping systems would be higher than that of monoculture systems.

## MATERIALS & METHODS

### Experimental site description and design

The study was conducted at a pecan orchard in Wenji in the Yingquan District of Fuyang City, Anhui Province, China (33°3′N, 115°36′E). This area has a warm temperate semi-humid monsoon climate with an annual average temperature of 14.9 °C and an annual average precipitation of 889 mm. The maximum temperature for this area was 41.4 °C and the minimum temperature was −20.4 °C (China Meteorological Data Service, http://data.cma.cn/).

The afforestation of the experimental site occurred in 2016 using the 'Pawnee' variety of tree in the second year of growth. Rows were spaced 4.0 m × 6.0 m. Three types of perennials (*Paeonia suffruticosa*, *Hemerocallis citrina*, and *Paeonia lactiflora*) were planted in the pecan plantation in 2017. A randomized block design was adopted for the study in September 2019 consisting of four treatments and three replicates. The treatments included: (1) PPH, with a row spacing of *Paeonia suffruticosa* of 0.2 m × 0.2 m and *Hemerocallis citrina* spaced at 0.4 m × 0.8 m; (2) PPS, with the row spacing of *Paeonia suffruticosa* at 0.2 m × 0.6 m; (3) PPL, with the row spacing of *Paeonia lactiflora* at 0.2 m × 0.6 m; (4) pecan monoculture (CK).

### Soil sampling

Soil samples were collected from the experimental site in September 2019. Eight pecan seedlings represented one plot, and three plots were randomly established for each treatment in the selected sampling area. A shovel was used to remove the plants and the surface litter from the soil. Five random soil profiles were obtained using a zig-zag sampling pattern (*Tafesa, Chimdi & Aga, 2019*)) at depths of 0–20 cm, 20–40 cm, and 40–60 cm. The samples were then mixed to generate a soil sample for each layer. A total of 36 soil samples were collected from the four treatment sites. The samples were sealed in plastic bags and transported to the laboratory. Soil samples were dried to determine their physicochemical properties and were kept in refrigerator at 4° for determining enzyme activities.

### Soil physicochemical properties analysis

Soil samples were collected using the ring knife method to determine their moisture content (MC), bulk density (BD), and porosity (TPO) (*Pan et al., 2017*). Following the removal of impurities, the $NO_3^-$-N, $NH_4^+$-N, AP, AK, pH value, electrical conductivity value (EC), total phosphorus (TP), total potassium (TK), total carbon (TC), total nitrogen (TN), available potassium (AK), Ca and Mg contents were determined. The soil $NO_3^-$-N, $NH_4^+$-N, TP, AP and AK were measured using an automatic discontinuous chemical analyzer (CleverChem Anna, Germany) (*Si et al., 2018*; *Yang et al., 2021*). The pH value of the soil was determined using a pH meter (Mettler Toledo, Shanghai, China) in a 1:2.5 (w/v) soil solution (*Ma et al., 2021*). The EC value of the soil was determined using an electrical conductivity meter in a soil-water extract at 1:5 at 25° (*Xie et al., 2019*). The TC and TN of the soil were determined *via* an automatic element analyzer (Vario EL Cube, Germany Elementar) (*Wang et al., 2018*). The contents of TK, Ca, and Mg in the soil were

measured using an inductively coupled plasma emission spectrometer (iCAP 6300 Series, America ThermoFisher) (*Ma et al., 2019*).

## Soil enzyme activity

The fresh soil samples from the surface layer (0–20 cm) were air-dried and sifted through a 50 mesh. The activity of seven types of soil enzymes, including S-POD, S-PPO, S-UE, S-AKP, S-BG, S-CBH, and S-NAG, were studied. S-AKP activity was determined by the p-nitrophenylphosphonate disodium method and S-POD activity was determined by potassium permanganate titration. S-UE activity was determined by the sodium phenol-sodium hypochlorite colorimetric method; S-CBH activity was determined by the phenyldisodium phosphate colorimetric method; S-NAG activity was determined by the p-nitrophenol colorimetric method; S-PPO activity was determined by the pyrogallic acid colorimetric method; and S-BG activity was determined by the microporous plate fluorescence substrate method (*Domínguez et al., 2017*; *Weintraub et al., 2013*). The international unit of enzyme consumption per gram of soil (U/g) was used as the unit of measure.

## Statistical analysis

SPSS 19.0 and Origin Pro 2021 software were used to analyze the comprehensive data. The data derived from the different soil depths (0–20 cm, 20–40 cm, and 40–60 cm) were statistically analyzed by single-factor analysis of variance (ANOVA), whereas significant differences ($p < 0.05$) in the physical and chemical properties of the soils of the various agroforestry systems were evaluated using a minimum significant difference test (LSD). Correlation analysis was employed to examine the relationships between the physical and chemical properties of the different soil layers, as well as between the physical and chemical properties and enzyme activity in the topsoil (0–20 cm). The PCA ranking performed in Origin Pro 2021 was used to analyze the physical and chemical properties of the soil.

## RESULTS

### Soil physical properties

The basic physical properties of the soil samples extracted from different soil depths and agroforestry patterns of pecan, including the EC, MC, BD, and TPO are shown in Fig. 1. In the 0–20 cm soil layer, the EC of PPL was significantly higher than that of the PPS and CK ($p < 0.05$). The soil MC of the PPS group was significantly higher than that of the CK ($p < 0.05$). The soil BD of all the soil samples ranged 1.42 g/cm$^3$ to 1.55 g/cm$^3$. The soil BD of the PPS and CK was significantly higher than that of the PPH and PPL ($p < 0.05$).

In the 20–40 cm soil layer, the EC of the CK group soil was the lowest (Fig. 1). The MC of the soil samples from the CK and PPS groups was significantly higher than that of the PPH and PPL groups ($p < 0.05$). The BD of all the soil samples ranged from 1.39 g/cm$^3$ to 1.71 g/cm$^3$, and the value of BD in descending order was PPS, CK, PPL, and PPH, where there was a significant difference between any two ($p < 0.05$). The TPO under PPH group was significantly higher than that of other groups ($p < 0.05$).

In the 40–60 cm soil layer, the EC of the PPL soil was significantly higher than that of the CK, PPH, and PPS ($p < 0.05$). The soil MC of the PPS group was significantly higher than

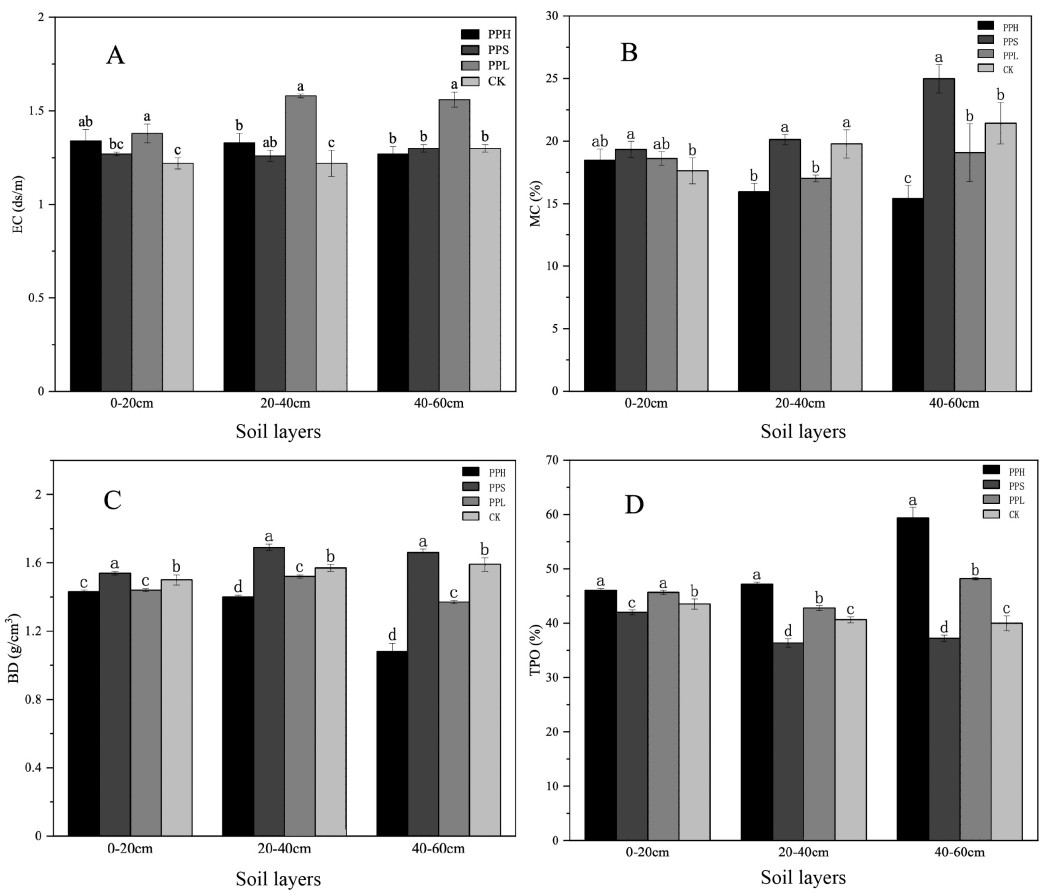

**Figure 1** **Soil physical properties under different soil layers and different agroforestry patterns.** (A) Changes of soil EC value; (B) changes of soil MC; (C) changes of soil BD; (D) changes of soil TPO. There were significant differences in one-way ANOVA of different compound patterns with different letters (LSD, $P < 0.05$).

that of the CK group ($p < 0.05$). The difference in the soil MC between the PPH and PPS groups was significant ($p < 0.05$). The BD of all the soil samples ranged from 1.02 g/cm$^3$ to 1.68 g/cm$^3$, and the BD of the all three soil layers, in all modes, was PPS > CK > PPL > PPH.

## Soil chemical properties

There were some variations in the chemical properties of the different agroforestry systems in the same soil layer (Table 1 and Fig. 2). Compared with the CK group, the agroforestry system increased the content of some elements in the soil. There were significant differences in the $NO_3^-$-N, TN, AK, TK, and TC contents between the four pecan agroforestry systems. The average pH value of all soil layer samples ranged from 7.80 to 8.40, and the pH decreased by 0.96% - 2.41% in contrast to the monoculture. The content of Mg in the PPS group was significantly higher than that in the PPH, PPL, and CK groups in all three soil layers ($p < 0.05$).

Moreover, the TP and TN contents in the 0–20 cm soil layer of the CK group were significantly lower than those of the PPS and PPL groups ($p < 0.05$). The contents of

Wang et al. (2022), *PeerJ*, DOI 10.7717/peerj.12663

**Table 1  Basic chemical properties of soils in various agroforestry systems in different soil layers.** The lowercase letters of different agroforestry systems in the same soil layer were different, and the difference was significant ($P < 0.05$).

| Treatments | AP (mg/kg) | TP (g/kg) | $NH_4^+$-N (mg/kg) | $NO_3^-$-N (mg/kg) | TN (g/kg) | AK (g/kg) | TK (g/kg) | TC |
|---|---|---|---|---|---|---|---|---|
| 0–20 cm | | | | | | | | |
| PPH | 9.99 ± 2.13a | 1.72 ± 0.14ab | 0.69 ± 0.36a | 2.34 ± 1.21b | 0.94 ± 0.12a | 244.27 ± 55.22a | 3.18 ± 0.42b | 16.40 ± 1.62a |
| PPS | 10.42 ± 1.47a | 1.98 ± 0.37a | 0.61 ± 0.52a | 1.76 ± 0.78b | 0.94 ± 0.04a | 223.70 ± 61.20a | 4.15 ± 0.33ab | 16.54 ± 0.42a |
| PPL | 8.43 ± 1.81a | 1.84 ± 0.23a | 0.38 ± 0.14a | 6.08 ± 2.28a | 0.87 ± 0.03a | 78.37 ± 25.26b | 4.47 ± 0.79a | 14.70 ± 0.10ab |
| CK | 6.76 ± 3.21a | 1.29 ± 0.26b | 0.14 ± 0.11a | 2.52 ± 0.84b | 0.48 ± 0.25b | 52.93 ± 7.90b | 4.34 ± 0.80ab | 13.63 ± 1.00b |
| 20–40 cm | | | | | | | | |
| PPH | 3.58 ± 1.73a | 0.98 ± 0.02a | 0.89 ± 0.35a | 2.82 ± 1.92b | 0.35 ± 0.03b | 149.20 ± 42.52a | 2.51 ± 0.31c | 12.09 ± 0.16b |
| PPS | 2.19 ± 0.90a | 1.15 ± 0.04a | 0.32 ± 0.04b | 0.57 ± 0.08b | 0.60 ± 0.05a | 147.90 ± 11.89a | 4.89 ± 0.35a | 16.44 ± 0.73a |
| PPL | 3.05 ± 2.73a | 1.12 ± 0.16a | 0.47 ± 0.15b | 7.85 ± 3.12a | 0.42 ± 0.11b | 119.67 ± 27.14a | 4.04 ± 0.67b | 13.17 ± 0.61b |
| CK | 1.51 ± 0.25a | 1.01 ± 0.08a | 0.21 ± 0.12b | 2.57 ± 1.31b | 0.29 ± 0.06b | 28.43 ± 3.76b | 3.73 ± 0.23b | 12.79 ± 0.80b |
| 40–60 cm | | | | | | | | |
| PPH | 2.25 ± 1.07a | 0.96 ± 0.07a | 0.60 ± 0.10a | 2.14 ± 0.96b | 0.21 ± 0.04b | 97.60 ± 40.45ab | 2.17 ± 0.48b | 11.39 ± 0.19b |
| PPS | 1.25 ± 0.46a | 1.10 ± 0.06a | 0.29 ± 0.20b | 0.56 ± 0.25b | 0.37 ± 0.06a | 113.27 ± 23.08a | 4.76 ± 1.25a | 19.68 ± 0.62a |
| PPL | 1.32 ± 0.08a | 1.07 ± 0.22a | 0.28 ± 0.10b | 9.76 ± 2.33a | 0.25 ± 0.11ab | 58.10 ± 25.75bc | 3.52 ± 0.95ab | 12.52 ± 1.98b |
| CK | 2.58 ± 2.85a | 0.92 ± 0.05a | 0.16 ± 0.14b | 2.28 ± 1.00b | 0.18 ± 0.06b | 16.50 ± 5.93c | 3.36 ± 0.26ab | 10.85 ± 0.18b |

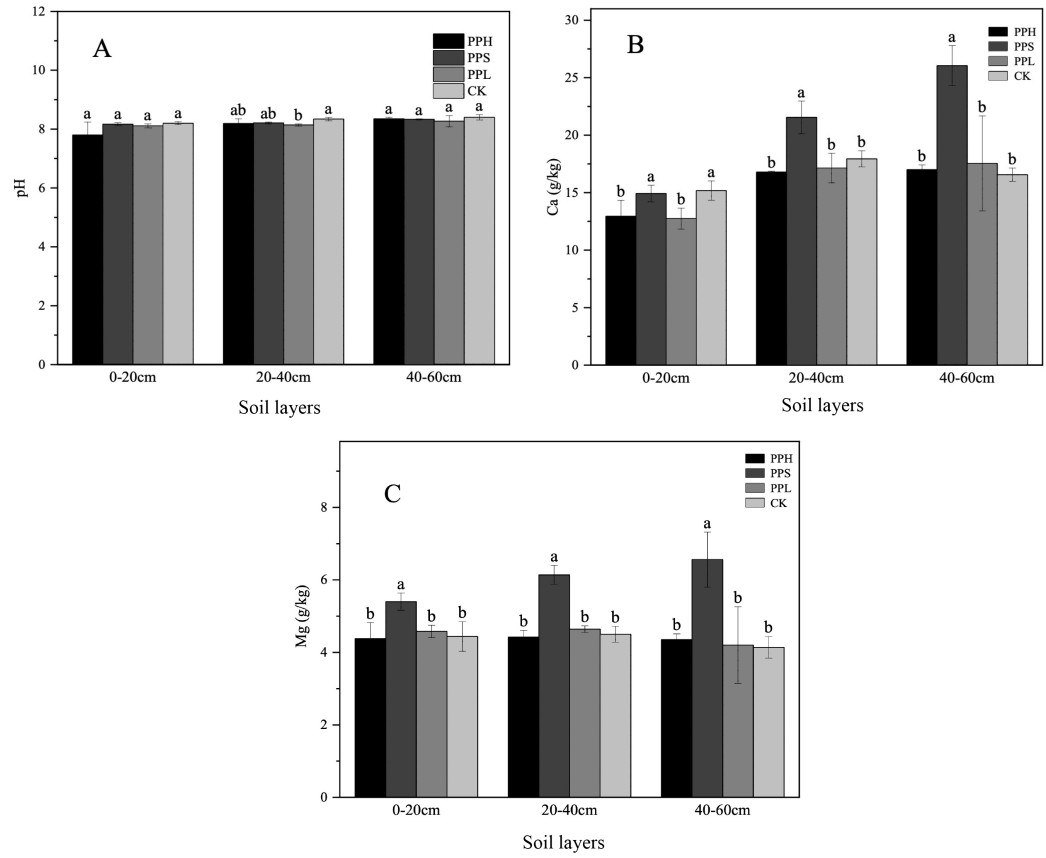

**Figure 2** **Soil pH (A), Ca (B), Mg (C) in different soil layers under various agroforestry systems.** The lowercase letters of different agroforestry systems in the same soil layer were different, and the difference was significant ($P < 0.05$).

AP, $NH_4^+$-N, AK, and TC in the CK group were lower than that of the soil following agroforestry. There were significant differences in the soil $NO_3^-$-N between the PPL and the PPH, PPS, and CK groups ($p < 0.05$). The TK content of the PPS group soil was significantly higher than that of the PPH, PPL, and CK groups ($p < 0.05$).

### Soil enzyme activity

Significant effects of different agroforestry groups on the S-UE, S-AKP, S-AKP, S-CBH, S-NAG, S-POD and S-BG were observed (Table 2). Intercropping significantly increased the activity of S-AKP in soil compared to CK ($p < 0.05$). The activity of S-POD and S-BG in the PPL were significantly lower than those in the CK group ($p < 0.05$). The activity of S-CBH in the PPH and PPS was significantly higher than in PPL and CK ($p < 0.05$). However, the activity of S-NAG was opposite and was significantly higher in PPL and CK than in the PPH and PPS ($p < 0.05$).

**Table 2 Activities of seven main enzymes in the topsoil of different pecan agroforestry systems.**

| Treatment | S-UE (U/g) | S-AKP (U/g) | S-PPO (U/g) | S-CBH (U/g) | S-NAG (U/g) | S-POD (U/g) | S-BG (U/g) |
|---|---|---|---|---|---|---|---|
| PPH | 893.89 ± 15.12bc | 10.77 ± 0.57a | 12.22 ± 0.62a | 4.46 ± 0.46b | 6.59 ± 1.38a | 4.60 ± 0.99ab | 129.58 ± 29.45a |
| PPS | 1018.64 ± 67.09a | 11.72 ± 0.53a | 9.76 ± 0.43b | 5.43 ± 0.91b | 7.14 ± 0.38a | 4.91 ± 0.76ab | 131.77 ± 19.53a |
| PPL | 974.79 ± 54.41ab | 10.86 ± 0.98a | 10.85 ± 0.63ab | 11.18 ± 2.83a | 3.13 ± 0.79b | 3.38 ± 1.16b | 24.84 ± 6.50b |
| CK | 831.98 ± 33.33c | 8.48 ± 0.49b | 11.50 ± 1.39a | 12.52 ± 2.12a | 2.39 ± 0.58b | 5.35 ± 0.50a | 150.84 ± 16.58a |

Notes.

S-UE, urease; S-AKP, alkalinephosphatase; S-PPO, polyphenoloxidase; S-CBH, cellobiohydrolase; S-NAG, 1,4- $\beta$-N-acetylglucosaminidase; S-POD, peroxidase; S-BG, $\beta$-1,4-glucosidase.

The lowercase letters of different agroforestry systems in the same soil layer were different, and the difference was significant ($P < 0.05$).

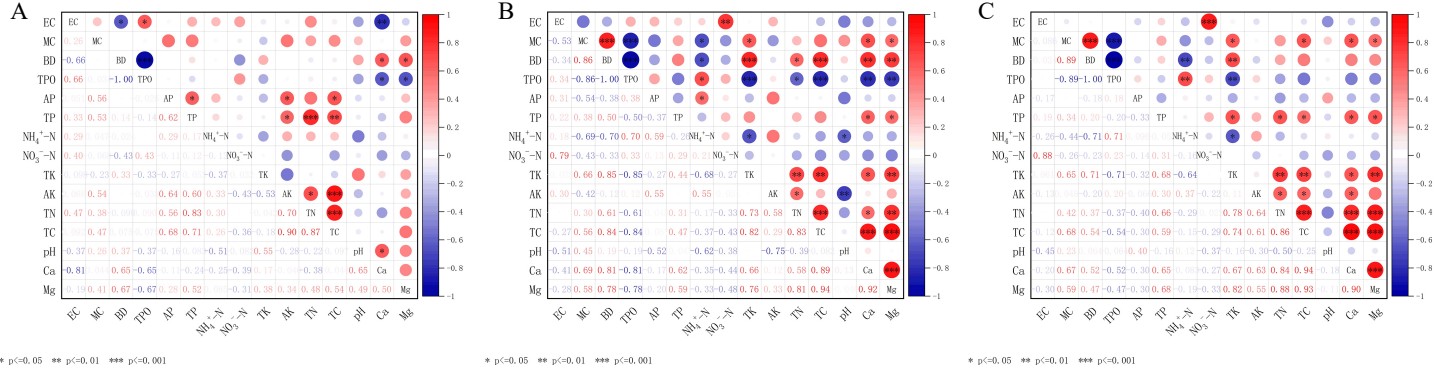

**Figure 3 Correlation matrix between soil physical and chemical properties in different soil layers.** Correlation matrix between physical and chemical properties of 0–20 cm soil (A); 20–40 cm correlation matrix between soil physical and chemical properties (B); 40–60 cm correlation matrix between soil physical and chemical properties (C). S-UE, urease; S-AKP, alkalinephosphatase; S-PPO, polyphenoloxidase; S-CBH, cellobiohydrolase; S-NAG, 1,4-β-N-acetylglucosaminidase; S-POD, peroxidase; S-BG, β-1,4-glucosidase.

## Correlations between physicochemical properties and enzyme activity in different layers of soil

Fig. 3 summarizes the correlations between the main physical and chemical properties of the treatments in the different soil layers (0–20 cm, 20–40 cm, and 40–60 cm). The results revealed that in the 0–20 cm soil layer, the EC was negatively correlated with the BD and Ca content ($p < 0.05$), and positively correlated with the TPO ($p < 0.05$). Further, the BD was positively correlated with the Ca and Mg contents ($p < 0.001$). The AP was positively correlated with the TP, AK, and TC ($p < 0.05$). The TP and AK were positively correlated with the available TN and TC ($p < 0.05$). There was a significantly positive correlation between the TN and TC ($p < 0.001$), as well as between the pH and total carbon Ca content ($p < 0.05$).

In the 20–40 cm and 40–60 cm soil layers, there was a strong significant positive correlation between the soil EC and $NO_3^-$-N ($p < 0.001$). The soil MC and BD were significantly negatively correlated with the TPO ($p < 0.001$), and positively correlated with the TK ($p < 0.05$). The soil BD was negatively correlated with the $NH_4^+$-N and positively correlated with the TK ($p < 0.05$), whereas the TP was positively correlated with the Ca and Mg contents ($p < 0.05$). There was a significantly positive correlation among the TK

**Table 3** Pearson correlation coefficient between soil physical and chemical properties and soil enzyme activities in 0–20 cm soil layer.

|  | S-UE | S-AKP | S-PPO | S-CBH | S-NAG | S-POD | S-BG |
|---|---|---|---|---|---|---|---|
| EC | 0.322 | 0.411 | 0.182 | −0.150 | 0.047 | −0.681[*] | −0.648[*] |
| MC | 0.530 | 0.634[*] | −0.265 | −0.284 | 0.503 | 0.180 | −0.292 |
| BD | 0.180 | 0.031 | −0.653[*] | −0.043 | 0.156 | 0.357 | 0.484 |
| TPO | −0.180 | −0.032 | 0.654[*] | 0.043 | −0.156 | −0.356 | −0.483 |
| AP | 0.415 | 0.466 | −0.345 | −0.305 | 0.732[**] | 0.158 | −0.084 |
| TP | 0.432 | 0.691[*] | −0.643[*] | −0.357 | 0.512 | −0.050 | −0.358 |
| $NH_4^+$-N | 0.495 | 0.425 | 0.207 | −0.678[*] | 0.572 | −0.410 | 0.032 |
| $NO_3^-$-N | 0.062 | 0.009 | −0.031 | 0.461 | −0.423 | −0.383 | −0.709[*] |
| TK | 0.241 | −0.164 | −0.487 | 0.457 | −0.441 | −0.312 | −0.234 |
| AK | 0.194 | 0.621[*] | −0.147 | −0.814[**] | 0.874[***] | 0.315 | 0.245 |
| TN | 0.471 | 0.749[**] | −0.451 | −0.596[*] | 0.617[*] | −0.259 | −0.269 |
| TC | 0.364 | 0.677[*] | −0.375 | −0.730[**] | 0.799[**] | 0.132 | 0.053 |
| pH | 0.091 | −0.050 | −0.310 | 0.354 | −0.390 | 0.236 | −0.208 |
| Ca | −0.100 | −0.327 | −0.207 | 0.082 | −0.034 | 0.699[*] | 0.447 |
| Mg | 0.560 | 0.427 | −0.751[**] | −0.319 | 0.396 | 0.146 | 0.046 |

**Notes.**

[*]Significance is at $p < 0.05$.

[**]Significance is at $p < 0.01$.

[***]Significance is at $p < 0.001$.

Soil physical and chemical properties include EC, MC, BD, TPO, AP, TP, NH4+- N, NO3–N, TK, AK, TN, TC, soil pH, Ca and Mg. Soil enzyme activities include S-UE, S-AKP, S-PPO, S-CBH, S-NAG, S-POD, S-BG.

and TN, TC, Ca, and Mg contents ($p < 0.05$), as well as between the AK and TN ($p < 0.05$). Further, there was a very significant positive correlation among contents of TC and TN, Ca and Mg ($p < 0.001$), and an extremely significant positive correlation between contents the Ca and Mg ($p < 0.001$).

The enzyme activity was affected by the physical and chemical properties of the soil (Table 3). The activities of S-NAG and S-AKP were significantly positively correlated with AK, TN and TC, while the activity of S-CBH showed a significantly negative relationship with AK, TN and TC ($p < 0.05$). The activities of S-UE, S-POD and S-BG exhibited a relatively weak correlation with the physical and chemical properties of the soil, and TK and pH had no significant correlation with all of the soil enzyme activities.

## Principal component analysis of soil physical and chemical properties

Table 4 showed the weights of the 15 original variables along with the first four principal components. According to the PCA ranking results, the eigenvalues of the first four ranking axes were greater than 1, and the cumulative contribution rate reached 81.55% (Fig. 4, Table 3). The main axis (PC1) contributed 32.93% of the total variance, the second principal component (PC2) explained 28.09% of the total variance, the third principal component (PC3) contributed 11.55% to the total variance, and the fourth principal component (PC4) contributed 8.98% to the total variance.

**Table 4** Loads and explained variances of 15 original variables in the first four principal components in principal component analysis (PCA).

| Soil Physico-Chemical Properties | Principal Components | | | |
|---|---|---|---|---|
| | PC1 | PC2 | PC3 | PC4 |
| Soil electrical conductivity(EC) | −0.1388 | 0.0051 | 0.5871 | 0.3834 |
| Soil water content (MC) | 0.3282 | −0.1627 | 0.0525 | −0.0141 |
| Soil bulk density(BD) | 0.3717 | −0.0857 | 0.1724 | −0.1318 |
| Total porosity of soil(TPO) | −0.3717 | 0.0857 | −0.1725 | 0.1318 |
| Soil available phosphorus (AP) | 0.0319 | 0.4278 | −0.0152 | −0.248 |
| Soil total phosphorus (TP) | 0.1311 | 0.4082 | 0.1195 | −0.156 |
| Soil ammonium nitrogen($NH_4^+$-N) | −0.1455 | 0.2207 | −0.2497 | 0.4469 |
| Soil nitrate nitrogen($NO_3^-$-N) | −0.194 | 0.0032 | 0.6078 | 0.1857 |
| Soil total potassium (TK) | 0.3541 | −0.0017 | 0.2662 | −0.1226 |
| Soil available potassium (AK) | 0.1063 | 0.3629 | −0.1684 | 0.3398 |
| Soil total nitrogen (TN) | 0.1761 | 0.4272 | 0.0678 | −0.093 |
| Total soil carbon(TC) | 0.3804 | 0.1608 | −0.041 | 0.2386 |
| Soil pH (pH) | 0.0116 | −0.3615 | −0.1302 | −0.1667 |
| Ca | 0.2464 | −0.2976 | −0.1155 | 0.4309 |
| Mg | 0.3827 | −0.0353 | −0.1053 | 0.2977 |
| Eigenvalue | 4.9399 | 4.2136 | 1.733 | 1.3464 |
| Percentage of Variance (%) | 32.93 | 28.09 | 11.55 | 8.98 |
| Cumulative (%) | 32.93 | 61.02 | 72.58 | 81.55 |

## DISCUSSION

### The effect of agroforestry systems on the physicochemical properties of soil

Pecan agroforestry groups (PPH and PPL) were shown to reduce the BD, increase the MC, and increase the TPO of the soil when compared with CK,. These results may be attributed to leaf litter introduced by the vegetation and the distribution and penetration of the roots into the soil to improve the physical properties and structure of soil (*Chen et al., 2019*; *Stöcker et al., 2020*). It has been reported that soil BD increases with the soil depth (*Stöcker et al., 2020*). Our results showed decreased BD in the deep soil of the PPH group, which is agreement with *Ling et al. (2020)*, and requires further investigation. Due to the decomposition of litter and chemical degradation of minerals, the soil EC level increased following agroforestry (*Samani et al., 2020*). In this study, the EC levels in the surface layer (0–20 cm) and subsurface layer (20–40 cm) of the soil increased after the agroforestry of pecans. The EC levels decreased in the deeper soil layer (40–60 cm) after agroforestry compared with the CK, which may be related to the distribution of the root systems and biochemical cycles of vegetation (*Pierret et al., 2016*). The vertical roots of pecan plants are concentrated below 40 cm (*Xu et al., 2019*) and their taproot depth is much greater than that of intercropped plants (*Hanson, 2019*).

Earlier studies have documented the use of agricultural intercropping to improve soil fertility (*Du et al., 2019*; *Xia et al., 2019*). We found that agroforestry increased the

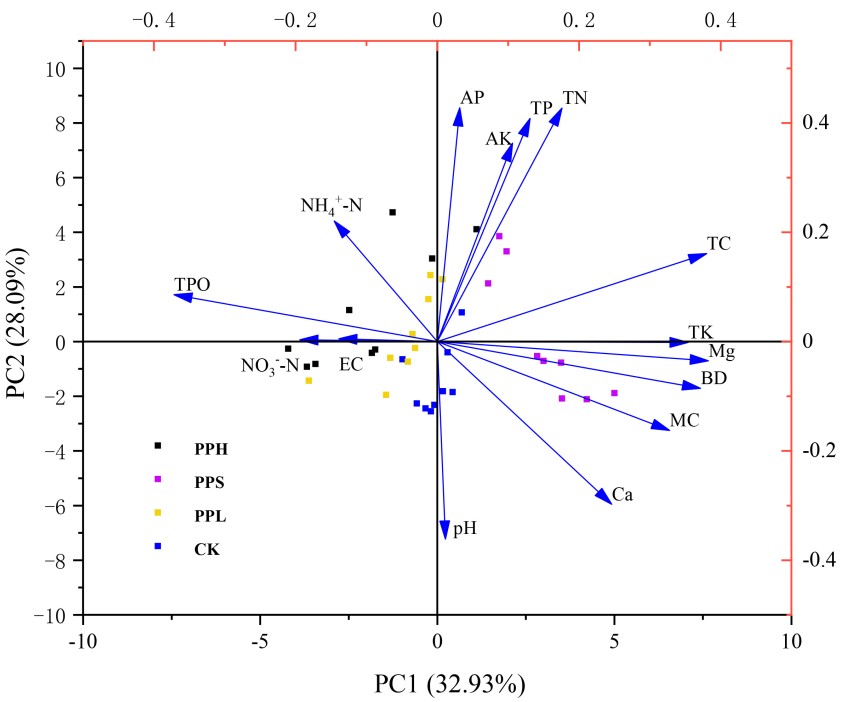

**Figure 4   PCA ranking chart of soil physical and chemical properties of different pecan agroforestry models.** Blue arrows indicate soil physical and chemical factors. Plots are represented by different color symbols. Specifically, black squares indicate PPH; red squares represent PPS; green squares represent PPL; blue squares represent CK. The abbreviations of soil physical and chemical properties are shown in Fig. 2. There was a significant correlation between the angle of intersection and its corresponding variable, where an acute angle represents a positive correlation, an obtuse angle represents a negative correlation, and a right angle represents an insignificant correlation.

nutrient content, improved nutrient utilization, and promoted nutrient cycling in the soil (*Mortimer, Saj & David, 2018*). These results may be attributed to other mechanisms, such as changes in community composition and biomass, organic matter inputs, and the microclimate or soil structure (*Borden, Thomas & Isaac, 2019*; *Wu et al., 2020*). Most of the soil nutrients (*e.g.*, AP, TP, $NH_4^+$-N, AK, TN, TC) measured in the soil of the pecan agroforestry groups (PPH, PPS, and PPL) and pecan monoculture (CK) during this study were increased in the surface soil. Additionally, PPS increased the contents of Ca and Mg at the average soil depth. Therefore, pecan intercropping was shown to improve the availability of soil nutrients to a certain extent.

Changes in the physicochemical properties of soil caused by intercropping may be due to differences in the distribution of plant-soil roots and litter cover in the pecan agroforestry systems (*Cardinael et al., 2020*; *Li et al., 2016*). The pH of the soil may affect the acid–base balance of microbial cells and regulate the utilization of soil nutrients (*Kemmitt et al., 2006*). In our study, the soil pH was lower than that of the pecan monocropping after planting crops in the pecan plantation, indicating that agroforestry may improve the soil pH, and maintain soil productivity. These findings were congruent with the findings of *Hu et al. (2019)*. Key biological functions of plant roots such as water and nutrient uptake,

respiraton, and exudation alter pH in the soil *Rudolph et al. (2013)*. Our data revealed that the soil pH was lower at the surface layer (0–20 cm) than the deeper soil layers. This effect may be attributed to the shallower root distribution of *P. suffruticosa* and *H. citrina*, which was consistent with the results of *Andrianarisoa et al. (2016)*. The pH value of topsoil decreased after intercropping, which prevented excessive soil alkalization.

## Different effects of agricultural and tree intercropping on enzyme activity in the surface soil of pecan plantation

Enzyme activity is a critical index of soil fertility and quality and is important in the soil's biochemical function (*Nannipieri, Trasar-Cepeda & Dick, 2017*). We found that some of the enzyme activity from pecan cultivation was significantly higher than that of the single cultivation of pecans. These results were consistent with the findings of other intercropping systems (*Clivot et al., 2019*; *Ma et al., 2017*). Thus, a better understanding of these seven enzymes can clarify the role of agroforestry systems to improve soil fertility. Soil S-UE, S-BG, S-CBH, S-NAG, and S-AKP are all hydrolases. Among them, S-BG and S-CBH are involved in the C cycle, S-UE and S-NAG are involved in the N cycle, and S-AKP participates in the P cycle (*Adetunji et al., 2017*; *Li et al., 2019*). In the present study, S-POD and S-PPO were oxidoreductases, and oxidoreductase was involved in the synthesis of soil humus components and in the process of soil formation, which helps to understand the nature of soil occurrence and related soil fertility (*Ananbeh et al., 2019*).

Our research found that in the surface soil, the S-UE, S-AKP, and S-NAG activity of the soil were significantly increased under the agroforestry system when compared to CK. However, the S-BG and S-POD activity of PPL intercropping significantly declined, when compared with the CK, which may be attributed to the fact that soil moisture affects the biochemical process of soil carbon conversion catalyzed by S-BG (*Zhang et al., 2011*). The S-BG enzyme activity is reduced when soil moisture decreases, which reduces the nutrient renewal speed and lowers the supply of plant nutrients (*Adetunji et al., 2017*). Compared with CK, the S-PPO and S-CBH activity significantly decreased in soils under PPS intercropping. This may be due to the competition and interaction between different species under different compound management modes (particularly root system and root exudates), which affected crop water and nutrient absorption (*Karaca et al., 2010*).

The activity of surface soil enzymes were found to be closely related to the distribution of soil C, N, and P in various systems, which confirmed the key roles of these soil enzymes in carbon and nitrogen cycling in the ambient environment (*Philippot et al., 2013*). Contents of AK, TN and TC were highly correlated with the activity of S-AKP, S-CBH, and S-NAG, and had the strongest correlation with S-NAG enzyme activity, which better explained the changes in S-NAG enzyme activity. TP was highly positively correlated with S-AKP, which indicated that TP was the main factor that directly or indirectly influenced the activity of S-AKP, which may have been due to its positive feedback effect. Although the roles of some enzymes studied in the nutrient cycle were not evident, their ability to promote the decomposition of plant litter appeared to explain the increased content of these elements in the soil, which was expected (*Feng et al., 2019*).

## CONCLUSIONS

Our research investigated the changes of soil quality associated with the conversion of pecans from a single crop to agroforestry. The purpose of the study was to elucidate how pecans can benefit from intercropping in young plantations, particularly relating to the improvement of physicochemical properties and enzyme activity. The results revealed that compared with pecan monocropping, the agroforestry systems were beneficial for improving the physical properties of the soil and optimizing the soil structure. Moreover, the test results showed that intercropping had a certain effect on soil nutrients, improved nutrient utilization efficiency, and increased soil enzyme activity to promote soil TC, TN, and TP nutrient cycling. Therefore, these systems can be incorporated for sustainable soil management practices so that farmers can obtain the best use of resources with limited land. Our research results have significant implications for the development and management of pecan agroforestry systems. This study can facilitate the maintaining of balance in the agroforestry systems; however, it is also necessary to conduct further in-depth studies on the root distribution and enzyme activity of intercropping plants, to correlate their changes with microbial composition, while understanding their regulatory mechanisms.

## ACKNOWLEDGEMENTS

The authors would like to thank Mingyuan Gu, Lei Wang, and Lei Zhao of the Anhui Agricultural University for their support in the collection of field data and soil processing.

### Funding

This research was supported by the Major Special Science and Technology Project of Anhui Province ( 202103b06020011 and 202103a06020007), and the Forestry Science and Technology Promotion Demonstration Project of State (2021TG 02), and Forestry Science and Technology Innovation Platform Project of State(2021-08). The funders had no role in study design, data collection and analysis, decision to publish, or preparation of the manuscript.

### Grant Disclosures

The following grant information was disclosed by the authors:
The Major Special Science and Technology Project of Anhui Province: 202103b06020011.
The Forestry Science and Technology Promotion Demonstration Project of State (2021TG 02).
Forestry Science and Technology Innovation Platform Project of State (2021-08).

### Competing Interests

The authors declare there are no competing interests. Haoxin Ge and Xiang Ge are employed by Fuyang Xinfeng Seed Industry Co., Ltd.

## Author Contributions

- Zhaocheng Wang performed the experiments, analyzed the data, prepared figures and/or tables, and approved the final draft.
- Mengyu Zhou, Hao xin Ge and Xiang Ge performed the experiments, prepared figures and/or tables, and approved the final draft.
- Hua Liu and Songling Fu conceived and designed the experiments, authored or reviewed drafts of the paper, and approved the final draft.
- Cheng Huang analyzed the data, prepared figures and/or tables, and approved the final draft.
- Yuhua Ma analyzed the data, authored or reviewed drafts of the paper, and approved the final draft.

## Data Availability

All the raw measurements are available in the Supplementary Files.

## Supplemental Information

Supplemental information for this article can be found online at http://dx.doi.org/10.7717/peerj.12663#supplemental-information.

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
