# Peer review of "Pecan agroforestry systems improve soil quality by stimulating enzyme activity"

_PeerJ, doi:10.7717/peerj.12663_

## Round 0.1 · original submission · Major Revisions

Dear Dr. Wang and co-authors,

I just received the reviews of your manuscript. Please, consider all comments and suggestions provided by both reviewers during the revision of your manuscript. Some issues need to be considered before the acceptance as follows.

1. The soil sample preparation for analysis needs modification.
2. The authors should follow the journal pattern thorough the whole manuscript.
3. The structure of the manuscript needs further improvement. For example, it needs a better summary about introduction, Materials and methods and results (too many paragraphs in these parts).

A comprehensive revision of the English and logic of the manuscript is necessary before submitting the new version.

Don't forget to include a letter response along with the revised version of the manuscript. In this letter you must respond point by point to each question.

Best regards,

Xiaoming Kang

Reviewer 1 ·

Basic reporting

The manuscript named “Pecan Agroforestry Systems Improve the Soil Quality by Stimulating the Enzyme Activity” was interesting to readers. In this manuscript, the English language is clear and relatively professional, but it still needs to be improved to ensure that an international audience can clearly understand your text. More importantly, there are some Chinese expressions in your manuscript. Some examples where the language could be improved include lines 51, 92-92, 218-219, 231-232 and so on. The current phrasing makes comprehension difficult. I suggest you have a colleague who is proficient in English and familiar with the subject matter review your manuscript, or contact a professional editing service.
Intro & background shows in context. Literature is relevant but the formation in text and part of references is relatively poor. For examples in line 53, 55, 66, 116 and so on. I suggest you should check the literatures in text and part of references carefully to promise the formation is correct and consistent.
The structure of manuscript conforms to Peerj standards. Figures are relevant and well labelled and described but not high quality because of poor definition. I suggest you can import figures as the .tif or .png.
Raw data supplied is complete.

Experimental design

This manuscript is within the scope of Peerj. Research questions were well defined, relevant and meaningful. It is stated how the research fills an identified knowledge gap. Rigorous investigation performed to a high technical & ethical standard. Methods described information to replicate but with insufficient detail in dealing samples and determination methods. The process of the field sampling and sample handling in lab were described not clear. In addition, the determination of soil enzyme activity is not detailed enough. Finally, there are no references about determination method of each index. I suggest you should add relevant literatures.

Validity of the findings

The findings in this manuscript was meaningful and instructive to some extent. All underlying data have been provided; they are robust, statistically sound. Conclusions are well stated, linked to original research question & limited to supporting results.

Additional comments

Abstract:
1. In background, I suggest you should rewrite because the current background and purpose are not related to research content. More importantly, “the formation of a mechanism that drives the organic combination of multi-industries” was not studied in manuscript, so I suggest you delete this sentence.
2. In results, there are some uncommon abbreviations which was not spelled out at first use, such as kinds of soil enzymes. Finally, the last sentence is suggested to be deleted because it has a poor correlation with previous contents and it do not belong to your results just literature review for other references.
Introduction:
1.Line 50: I suggest you should add some new and relevant references to ensure domestic and foreign research progress to be shown adequately.
2.Line 51:Repalce the ”maintenance” to “increase”.
3.Line53: Revise “to improve” to “to be improved”. The formation of “Q. S. Li et al., 2018” is wrong, please correct it. So was the Line 55, 66,116, 239.
4.Line63: There was no reference named “Chen et al., 2017”, please delete it or replace it.
Materials & Methods:
1.Line92-93: There are some grammatical mistakes. Foe example, there are some simple sentences without coordinating conjunctions. Please revise them.
2.Line103:In text, the “0~20cm” and “0-20cm” appear many times, but their formation do not keep consistent. Please revise them.
3.Line 110:The “water content (MC)” may is the “water content (WC)”. Please revise it.
4.Line133:the software that is used to the PCA analysis needs to be noted.
Results:
1.Line153:Please correct “the same layers” to “same layer”.
2.Line158: ”(p<0.05)” is “(p<0.05)”. In the following manuscript, there are many mistakes, such as “(p<0.05)” or “(P<0.05)”. Please revise them to “(p<0.05)”.
3.Line164: Please replace “Correlation study” to “Correlations”.
4.Line181:Please revise “between the PPS, PPL, and CK agroforestry systems” to “among PPS, PPL, CK”.
5.Line187: I suggest you can combine this paragraph and paragraph in line164.
Discussion:
1.Line218: Please revise the “Ca and Mg content” to “contents of Ca and Mg”.
2.Line231:Delete “furthermore” and revise the “and played” to “which plays”.
3.Line233:Please revise the “agroforestry cultivation of” to “with the cultivation of”.
Conclusions:
1.Line263: Please replace the “impacts on” to “changes of”.

Reviewer 2 ·

Basic reporting

This paper presents an interesting topic of the impact of pecan agroforestry systems on the soil quality by stimulating enzyme activity, which could help gain insights into optimized land utilization and the promotion of stand growth. The soil physical and chemical indicators of different treatments were measured, the correlations were analyzed between soil physicochemical properties and soil enzymes, the author must dig deeper into the innovation points and draw conclusions from their data. Another obvious problem with this paper is the lack of logic, many paragraphs are confusing and beyond comprehension.

Experimental design

no comment

Validity of the findings

no comment

Additional comments

Abstract:
Line 27-30,The abbreviation are unnecessary.
Line 32, 35 & 36, Please define “C”, “N”, “P”, “TP”, “TN”, “AK”, “TC”, “S-UE”, “S-AKP”, and “S-NAG” when they are first used. Please check such errors throughout the manuscript.
Line 32-36, Please reorganize your main conclusions to make them not so confused.
Line 39, “root interactions and the microbial compositions”. The conclusions are overstated. I suggest that you improve the description to provide more significance of your study.

Introduction:
Line 50. Delete “-”
Line 53, 55, 61 & 66, In-text citations should be corrected to consistent format according to the author guidelines. Please check such errors throughout the manuscript.
Line 61-62. An explanation of why the soil quality improved through soil enzymes should be provided.
Line 44-84 There is too much unnecessary information, while some important terms and information are left unexplained. Consider merging some paragraphs to make the introduction more focused

Materials & Methods
Line 103. Please leave a blank space between numbers and units.
Line 124. Please introduce the determination method of soil enzyme activity, not just the name of the company.
Line 127-133. Please pay attention to the paragraph format and carefully modify it according to the author guidelines.
Line 110. MC is the abbreviation of water content? Please provide abbreviate form in common use.

Results

Line 142 & Line 146. “(P < 0.05)”, Pay attention to the use of space symbols.
Line 157 Delete “from”.
Line 180-193. The authors should clearly present the results of statistical analyses, main effects and comparison between different treatments should also be reported clearly.

Discussion
Line 201 More discussion and interpretation on the soil quality affected by agroforestry system should be provided in comparison with the other agricultural soils.

References
Line 280. The references in this study should be updated according to checking the recent published literature.
Line 346 & 402. Before submitting a manuscript, be sure that the list of references is properly prepared and formatted.

Figures and Tables
1. The statistical analysis of Figure 3 was plainly unreasonable. Soil enzymes were measured at a depth of 0-20 cm, not include 20-40 and 40-60 cm.
2. Please add the title of X axis in the Figure 1, 2 & 3.
3. All the tables should be revised for the inconsistent format
4. Table 1, 2 & 3 are not necessary since the information is presented in Figure 1, 2 & 3.

Language
It is noted that your manuscript needs careful editing by someone with expertise in technical English editing paying particular attention to English grammar, spelling, and sentence structure so that the goals and results of the study are clear to the reader.

·

Basic reporting

Background, abstract and introduction part of the manuscript is well written as per the requirement of the topic.
Manuscript is structured as per the journal standards. Whereas, the main headings and the references given in running text of manuscript needs to follow journal pattern.
Avoid repetition of the text given in line number 75-80 (Introduction), which is related to the methodology, so it should be place under Materials & Methods heading.
Figures and tables have relevance to the research and are in required quality.
However, there have some sort of scope to improvement in language as comments give in sticky notes on the manuscript (pdf file). Minor corrections also have to incorporate in the manuscript as per comments given on pdf file of manuscript.
I thanks for the providing raw data, all figures and tables in separate files.

Experimental design

Original research is in the scope of the journal.
The topic of the research has scope in widely adoption of the agroforestry systems for conservation of natural resources especially soil quality and health.
Manuscript hypothesis were well designed by the authors and discussed well.
S sampling method: why this method is used for the study. Needs to give reference of that method.
Soil preparation method for analysing physico-chemical and enzyme activity needs to check and confirm it. (Soil samples were dried for determination of physicochemical properties and kept in refrigerator at 4° C for determining enzyme activities)
Methods used for the determination of soil physicochemical properties and enzyme activities as given with sufficient reference information.
Analysis of the data is done by using standard statistical tools.

Validity of the findings

The research includes very interesting idea and it has wide scope in further research such as belowground interactions between tree-crop components of the agroforestry systems.
The literature provided in the manuscript is sufficient and backing the current research.
The research is replicated well for minimizing the errors in outcomes of the study.
The data given in manuscript is sufficient, analysed by statistical tools as per requirements and nicely illustrated in the manuscript.
In 20-40 and 40-60 cm soil depth: When BD has positive correlation with Ca and Mg, then how MC is positively correlated with Ca and Mg? (Line 172-1174)
Use correct short forms of the different agroforestry systems given in manuscript. (Line 139)

Additional comments

Overall, the manuscript is well written and has wide scope as per journals requirement. It has further scope in advance research in belowground tree – crop interactions.
Manuscript has some major mistakes need to check and correct them for improving standard of the paper and journal.
Authors need to follow journal pattern in case of manuscript headings and reference style.
Manuscript needs revision as per comments given on pdf file.

---

## Round 0.2 · Minor Revisions

Dear Dr. Wang and co-authors,

Please, consider all comments and suggestions provided by both reviewers during the revision of your manuscript. A comprehensive grammar check of the manuscript is also needed before the acceptance.

Don't forget to include a letter response along with the revised version of the manuscript. In this letter you must respond point by point to each question.

Best regards,

Xiaoming Kang

Reviewer 1 ·

Basic reporting

The manuscript named “Pecan Agroforestry Systems Improve the Soil Quality by Stimulating the Enzyme Activity” has been resubmitted for publication in Peerj. This manuscript has really improved after revision. And authors replied to reviewers and included into the text most of these comments. However not all of them were considered in the manuscript.

Experimental design

After revision, research questions were better defined and relevant. Methods described information to replicate with more details in dealing samples and determination methods. The process of the field sampling and sample handling in lab were described more clearly. In addition, the determination of soil enzyme activity is detailed enough. Finally, there are some relevant references about determination method for each index. Overall, authors did a good work during the process of revision.

Validity of the findings

The findings in this manuscript was meaningful and instructive to some extent. All underlying data have been provided; they are robust, statistically sound. Conclusions are well stated, linked to original research question & limited to supporting results.

Additional comments

Abstract:
No comments.
Introduction:
1.Line 59: Please revised the “pecanS” to “pecans”.
Materials & Methods:
1.Line85: Delete the “,” in phase of “115 °36' E,”.
2.Line85: Please explain the meaning of “experences”. Maybe there is some spelling mistake for this word. Please amend it.
Results:
1. Line148: Please revised “(p<0.05)” to “(p<0.05)”. Statistical probability is p (small + italic). In the following manuscript, there are many mistakes, such as “(p<0.05)”. Please revise them to “(p<0.05, small + italic)”. Last time, the formation of “p” changed, after this comment was copied to system. So the italic “p” was not obvious.
2.Line185: Please replace “Correlation study” to “Correlations”.
3.Line 202-203: Please revised “TC and TN, Ca and Mg contents” and “the Ca and Mg contents” to “contents of TC and TN, Ca and Mg” and “contents of Ca and Mg”. Because “TC and TN, Ca and Mg contents” and “the Ca and Mg contents” are the translations according to Chinese. So was the “The AK, TN, and TC contents” in Line 283-284. Please correct it.
Discussion:
1. Line 252: Please add the “.” behind “the soil (Rudolph et al,. 2013)”.
2. Line 283: Delete the “,” behind the “TN”.
Conclusions:
No comments.

Annotated reviews are not available for download in order to protect the identity of reviewers who chose to remain anonymous.

Reviewer 2 ·

Basic reporting

The manuscript named “Pecan Agroforestry Systems Improve the Soil Quality by Stimulating the Enzyme Activity” presents an interesting topic and have a good writing. In the present status of the manuscript, I suggest a minor revision should be made before the publication in PeerJ.

Experimental design

The revised manuscript is within Aims and Scope of the journal. Research questions were well defined, relevant and meaningful. It is stated how the research fills an identified knowledge gap. Rigorous investigation performed to a high technical & ethical standard. In addition, I suggest you should add references or websites of meteorological data sources.

Validity of the findings

The findings in this manuscript was meaningful and instructive to some extent. All underlying data have been provided; they are robust, statistically sound. Conclusions are well stated, linked to original research question & limited to supporting results.

Additional comments

Introduction:
Line 58-69. The font size of this paragraph should be consistent.

Materials & Methods
Line 86-88. Add references or websites of meteorological data sources.

Results
Line 146. Delete “the”.
Line 185-186. Please simplify the headline.

Conclusions
Line 294-296. This section mainly introduces the important progress and significance of the research, please modify the unnecessary description.

---

## Round 0.3 · accepted · Accept

Dear authors,

I am pleased to inform you that, following the revision made based on the reviewer’s comments, your manuscript is now acceptable for publication in PeerJ.

Best regards

Xiaoming Kang